# Development of a Simple Kinetic Mathematical Model of Aggregation of Particles or Clustering of Receptors

**DOI:** 10.3390/life10060097

**Published:** 2020-06-26

**Authors:** Andrei K. Garzon Dasgupta, Alexey A. Martyanov, Aleksandra A. Filkova, Mikhail A. Panteleev, Anastasia N. Sveshnikova

**Affiliations:** 1Faculty of Physics, Lomonosov Moscow State University, 1/2 Leninskie gory, 119991 Moscow, Russia; garzon.ak15@physics.msu.ru (A.K.G.D.); aa.martyanov@physics.msu.ru (A.A.M.); aa.filjkova@physics.msu.ru (A.A.F.); mapanteleev@physics.msu.ru (M.A.P.); 2National Medical Research Centеr of Pediatric Hematology, Oncology and Immunology named after Dmitry Rogachev, 1 Samory Mashela St, 117198 Moscow, Russia; 3Institute for Biochemical Physics (IBCP), Russian Academy of Sciences (RAS), Russian Federation, Kosyigina 4, 119334 Moscow, Russia; 4Center for Theoretical Problems of Physico-Сhemical Pharmacology, Russian Academy of Sciences, 30 Srednyaya Kalitnikovskaya str., 109029 Moscow, Russia; 5Faculty of Biological and Medical Physics, Moscow Institute of Physics and Technology, 9 Institutskii per., 141700 Dolgoprudnyi, Russia; 6Department of Normal Physiology, Sechenov First Moscow State Medical University, 8/2 Trubetskaya St., 119991 Moscow, Russia

**Keywords:** computational modeling, particle aggregation, receptor clustering, Smoluchowski coagulation

## Abstract

The process of clustering of plasma membrane receptors in response to their agonist is the first step in signal transduction. The rate of the clustering process and the size of the clusters determine further cell responses. Here we aim to demonstrate that a simple 2-differential equation mathematical model is capable of quantitative description of the kinetics of 2D or 3D cluster formation in various processes. Three mathematical models based on mass action kinetics were considered and compared with each other by their ability to describe experimental data on GPVI or CR3 receptor clustering (2D) and albumin or platelet aggregation (3D) in response to activation. The models were able to successfully describe experimental data without losing accuracy after switching between complex and simple models. However, additional restrictions on parameter values are required to match a single set of parameters for the given experimental data. The extended clustering model captured several properties of the kinetics of cluster formation, such as the existence of only three typical steady states for this system: unclustered receptors, receptor dimers, and clusters. Therefore, a simple kinetic mass-action-law-based model could be utilized to adequately describe clustering in response to activation both in 2D and in 3D.

## 1. Introduction

The process of clustering of the receptors during cell activation is a crucial step for correct cell response in a plethora of signaling pathways [1,2,3]. Receptor clustering can be governed by both intracellular mechanisms (e.g., reorganization of the cytoskeleton [1]) and the changes in receptor conformation upon ligation [4], which results in the change in the affinity of the receptor molecules to each other [5]. An increase in the local concentration of the receptors upon their activation results in the amplification of the initial signal [6,7,8,9] by either an enhancement of the affinity of the receptors for the ligand [10,11,12,13] or the intracellular signaling proteins [14] or by other mechanisms [15,16,17].

The underlying mechanisms of the receptor clustering resemble particle aggregation or protein polymerization. The polymerization is one-dimensional in most cases [18,19], while the clustering of receptors on the membrane is two-dimensional [10,20,21]. Finally, an aggregation of particles or proteins in the solution is three-dimensional [18]. Therefore, a mathematical model of particle aggregation could be applied to receptor clustering or protein polymerization upon simplification and imposition of the dimensional restrictions.

Several mathematical models of the process of aggregation have been proposed [22,23,24,25,26,27,28]. One of the commonly used approaches is the cellular automata models [24], where the system is represented by a mesh of cells that have a finite number of states. Each cell evolves, according to a specific set of rules that represent the temporal evolution of the system. One of the latest automata models used to describe the aggregation process was proposed by Mukhopadhyay and De [29], where the authors developed a cellular automata model to study dynamics of a 2D-aggregation process of the disordered tissue cell population, based on the active motility and local reorganization of cells. The cellular automata approach may be one of the most accurate, but the mesh dimensionality limits it. For 3D space or in case of large number of particles, the mesh and, therefore, the computational cost of the model solution becomes unreasonable [28]. As a result, the cellular automata approach usefulness is limited to the investigation and description of the receptor clustering or the polymerization processes [30,31], while being impractical for the modeling of aggregation.

Another commonly used group of approaches is mass-action law-based models. In particular, the Smoluchowski coagulation–fragmentation equation-based models [22,23,26,27]. For example, Fornari et al. [23] developed a Smoluchowski-type model of spreading and propagation of aggregates of misfolded proteins in the brain, that takes into account aggregate transport through axons. The proposed model considers only monomer binding and, therefore, could not be generalized to a wide range of aggregation processes. A similar Smoluchowski-based approach for protein aggregation that considers conformational change of the native monomer (single protein), coagulation processes, and reversibility of smaller oligomers was described and investigated by Zidar et al. [32].

In our previous work, we introduced a simple mathematical model of platelet aggregation in stirred suspension [33]. Here we aim to demonstrate the applicability of this model to general clustering and aggregation/fragmentation processes. In this study, we utilize a Smoluchowski coagulation-fragmentation-based approach for mathematical modeling of these processes and compare the results with the 2-differential equation model from [33], where multi-particle aggregates/clusters are represented by one variable. To the best of our knowledge, this is the first model of general coagulation processes, that can be implicated as a part of more complex models.

## 2. Modeling Approaches

Two extended mass-action law derived Smoluchowski-based mathematical models (“Aggregation model” and “Clustering model”) were considered. The models consisted of sets of ordinary differential equations (ODEs). All of the following systems are systems of a finite number of particles. At the first moment, only free single particles are present in the system. The “2-equation model” was obtained upon the reduction of the number of the variables in the extended models.

### 2.1. Aggregation Model

#### 2.1.1. Model Equations

We consider aggregation to be a general binary Smoluchowski coagulation-fragmentation process, and it is represented by the following reaction: $A_{i}+A_{j}\;↔A_{i+j}$ for i, j ≥ 1, where $A_{i}$, $A_{j}$, $A_{i+j}$ denote the aggregates of sizes *i*, *j*, and *i* + *j*, correspondingly.

Let [*i*] (*t*) represent the concentration of aggregates of size *i* at time point *t*. It could be assumed that the system at initial time *t* = *0* is composed of a finite number of identical monomers (aggregates of size 1): $\left[1\right]\left(0\right)=p_{0}$, [i](0)=0 for i>1, which reversibly form aggregates of larger sizes. The rate of aggregation/fragmentation is proportional to the concentration of particles with specified rate constants. This results in the following Smoluchowski equations:(1)$$\{\begin{array}{c} \frac{d\left[i\right]}{dt}=\frac{1}{2}\overset{i-1}{\underset{j=1}{\sum }}(a_{i-j,j}\left[i-j\right]\left[j\right]-f_{i-j,j}\left[i\right])-\overset{\infty }{\underset{j=1}{\sum }}(a_{i,j}\left[i\right]\left[j\right]-f_{i,j}\left[i+j\right]),\;i>1\;\\ \frac{d\left[1\right]}{dt}=-\overset{\infty }{\underset{j=1}{\sum }}(a_{1,j}\left[1\right]\left[j\right]-f_{1,j}\left[1+j\right])\; \end{array}$$
where [*i*] denotes the concentration of aggregates of size *i; a_i,j_* is the rate constant of coagulation of particle with size *i* with the particle with size *j*; *f_i,j_* is the rate constant of fragmentation of an aggregate of size *i* + *j* into to particles of size *i* and *j*.

The parameters form matrixes or coagulation (*a_i,j_*) and fragmentation (*f_i,j_*) kernels in the terminology of Smoluchowski equations. We assume that there is no difference between the reactions $A_{i}+A_{j}\;↔A_{i+j}$ and $A_{j}+A_{i}\;↔A_{i+j}$, so kernels are considered to be symmetrical. In the real biological systems, aggregates cannot achieve an infinite size, so we restricted the maximal achievable cluster size to *N*. The values of *a_i,j_*, *f_i,j_*, and *N* are chosen based on the following assumptions:The particles could be considered as material points because of their low concentration and high local velocities. Therefore, the probability of the particles’ collision depends on their concentration and not on their sizes [32,33,34].The direct and reverse rate constants of a single particle attachment to aggregate or aggregate to another aggregate are different [33,35,36,37].The mechanism of fragmentation is assumed to be independent of the aggregate size.

Therefore, in the “Aggregation model,” we use the following kernels:$$a_{i,j}=\{\begin{array}{c} k_{2},\;i=1,\;j=1\\ k_{1},\;i=1,\;j>1\\ k_{-2},\;i>1,\;j>1 \end{array}\;\mathrm{and}\;f_{i,j}=\{\begin{array}{c} k_{-1},\;i=1,\;j\geq 1\\ k_{3},\;i>1,j>1 \end{array}.$$

The scheme of the model is presented in Figure 1a. The explicit system (1) becomes:(2)$$\{\begin{array}{c} \frac{d\left[1\right]}{dt}=-k_{1}\left[1\right]\overset{N-1}{\underset{i=2}{\sum }}[i]-2k_{2}{\left[1\right]}^{2}+k_{-1}\left(\overset{N}{\underset{i=2}{\sum }}\left[i\right]+\left[2\right]\right)\\ \begin{array}{ll} \frac{d\left[2\right]}{dt}= & k_{2}\left[1\right]\left[1\right]-k_{-2}\left[2\right]\left(\overset{N-2}{\underset{i=2}{\sum }}\left[i\right]+\left[2\right]\right)+k_{3}\left(\overset{N}{\underset{i=4}{\sum }}\left[i\right]+\left[4\right]-\left[2\right]\right)\\ & \;-k_{1}\left[1\right]\left[2\right]+k_{-1}\left(\left[3\right]-\left[2\right]\right) \end{array}\\ \begin{array}{ll} \frac{d\left[j\right]}{dt}= & k_{-2}\left(\overset{\frac{j}{2}}{\underset{i=2}{\sum }}\left[i\right]\left[j-i\right]-\left[j\right]\overset{N-j}{\underset{i=2}{\sum }}\left[i\right]-h_{j}\left[j\right]\right)+k_{3}\left(\overset{N}{\underset{i=j+2}{\sum }}\left[i\right]+h_{j}\left[2j\right]-\left([\frac{j}{2}]-1\right)\left[j\right]\right)\\ & \;-k_{1}\left[1\right]\left(\left[j\right]-\left[j-1\right]\right)+k_{-1}\left(\left[j+1\right]-\left[j\right]\right),\;\mathrm{for}\;j>2 \end{array} \end{array}$$
where $h_{j}=\{\begin{array}{c} 1,\;j\leq \frac{N}{2}\;\\ 0,\;j>\frac{N}{2}\; \end{array}$.

#### 2.1.2. Model Parameters

To describe the process of particle aggregation, we used experimental data of dithiothreitol (DTT)-induced bovine serum albumin (BSA) aggregation in the presence of arginine ethylester (ArgEE) [38]. In this dataset, the degree of BSA aggregation was estimated from the time-courses of the concentration of non-aggregated protein. Here we assumed the model variable [1] (single particles) to follow the experimental values.

The “Aggregation model” was also adapted to describe data on platelet aggregation in solution in response to activation by ADP [33]. Platelets are anuclear blood cells that prevent blood loss by upon blood vessel wall damage. The prime goal of platelets is to aggregate and thus to cover the injury in the vessel. The golden standard of the platelet function testing in clinics is the Light Transmission Aggregometry (LTA) [39,40,41], which is based on the measurement of optical density (OD) of stirred platelet suspension and represents platelet capability to aggregate. Experimental OD curves were approximated with OD function, that was constructed according to Beer-Lambert Law [42]. This is a physical law, which states that the decrease in intensity of light that passes through an absorbing medium of the length *l* is $\sim e^{-kl}$, where *k* is the absorbance index. The parameter *k*, in turn, is proportional to the concentration of particles in the medium and to the absorbance characteristics of the particle. As a result, we introduced the following function:(3)$$OD=\exp\left(\frac{\left[1\right]\left(1+c_{1}b\right)+\sum_{i=2}^{N}[\frac{i}{2}]\left[i\right]-p_{0}}{p_{0}}\right)$$
where *c*_1_*b* represents the initial increase in optical density that is observed in the experiment due to the platelet shape change [33]. We approximate it as $b=\frac{t}{t+lag}\to 1,\;c_{1}\approx 0.1\;$(where *t* denotes time from activation, *lag* denotes a lag-time parameter) to achieve precise description of data; however, this parameter rapidly approaches constant value and has no effect on aggregation after the first ten seconds. Lag-time does not have any relation with other constants; it is an independent parameter introduced to describe processes that are not related to aggregation (as platelet shape change). The experimental data of mean aggregate size was also obtained in the LTA assay and was estimated by the following function:(4)$$S\;\left(t\right)=\frac{\sum_{i=1}^{N}i\cdot \left[i\right]\left(t\right)}{\sum_{i=1}^{N}\left[i\right]\left(t\right)}$$

#### 2.1.3. Platelet Aggregation Experiments

Platelet aggregation was performed using Biola LA-230 turbi-diametric aggregometer. Healthy volunteers (*n* = 5), both men and women aged between 18 and 25 years were recruited into the study. Investigations were performed in accordance with the Declaration of Helsinki and approved by CTP PCP RAS ethics committee, and written informed consent was obtained from all donors. The aggregation protocol was similar to the described earlier [33]. Briefly, blood of healthy donors was collected into tubes containing Li-hirudin (SARSTEDT Monovette, Nümbrecht, Germany). For platelet rich plasma (PRP) preparation blood was centrifuged at 100× *g* for 8 min without brakes. The experiments were conducted in 200 μL aliquots of PRP with mixing by a stirrer at 800 rpm. Platelet poor plasma was prepared by centrifugation 2000× *g* 15 min of whole blood and then it was used as a reference. ADP (Sigma-Aldrich, St Louis, MO, USA) was added at various concentrations as the platelet activator. Before measurements, platelets were incubated at 37 °C. An optical signal was recorded every 1 s.

### 2.2. Clustering Model

#### 2.2.1. Model Equations

The equations for the “Clustering model” are derived from the system (1) with the following modification. We assume that the major difference between receptor clustering and particle aggregation is space dimensionality. Therefore, the probability of the collision of two large clusters is low. Thus, we can consider the attachment and detachment of receptor monomers or dimers only [20]. Therefore, the kernels of the system (1) will become the following:$$a_{i,j}=\{\begin{array}{c} \begin{array}{c} k_{1},\;i=1,\;j\geq 1\\ k_{2},\;i=2,\;j\geq 1 \end{array}\\ 0,\;i>2,\;j>2 \end{array}\;\mathrm{and}\;f_{i,j}=\{\begin{array}{c} k_{-1},\;i=1,\;j\geq 1\\ k_{-2},\;i=2,\;j\geq 1\\ 0,\;i>2,\;j>2 \end{array}$$

The “Clustering model” will be the following:(5)$$\{\begin{array}{c} \frac{d\left[1\right]}{dt}=-k_{1}\left[1\right]\overset{N-1}{\underset{i=1}{\sum }}[i]+k_{-1}\left(\overset{N}{\underset{i=2}{\sum }}\left[i\right]+\left[2\right]\right)-\;k_{2}\left[1\right]\left[2\right]+k_{-2}\left[3\right]\\ \frac{d\left[2\right]}{dt}=-k_{2}\left[2\right]\overset{N-2}{\underset{i=1}{\sum }}[i]+k_{-2}\left(\overset{N}{\underset{i=3}{\sum }}\left[i\right]+\left[4\right]\right)\;+\;k_{1}\left[1\right]\left(\left[1\right]-\left[2\right]\right)-k_{-1}\left(\left[2\right]-\left[3\right]\right)\\ \frac{d\left[j\right]}{dt}=k_{1}\left(\left[j-1\right]\left[1\right]-\left[j\right]\left[1\right]\right)+k_{-1}\left(\left[j+1\right]-\left[j\right]\right)+k_{2}\left(\left[j-2\right]\left[2\right]-\left[j\right]\left[2\right]\right)+k_{-2}\left(\left[j+2\right]-\left[j\right]\right),\;\mathrm{for}\;j\neq 1,\;2 \end{array}$$

The resulting model could be considered as an extended Becker–Döring model [43] with an addition of two reactions, which complicates the analytical solution.

#### 2.2.2. Model Parameters

Different experimental approaches can be used to observe the clustering process [44,45,46]. Fluorescent microscopy is the most attractive option as it allows us to directly visualize the process of receptor cluster formation [47,48]. To illustrate the ability of the “Clustering model” to describe experimental data, we used the data on glycoprotein VI (GPVI) receptor (platelet receptor for collagen [49,50]) clustering in response to different agonists [51], where clustering of fluorescently labeled GPVI receptors on the surface of single washed platelets was visualized by means of Total Internal Reflection Microscopy (TIRFM). The mean cluster size for the Col-III and III-30 (monovalent collagenous substrates that enhance GPVI clustering) was assessed from the experimental data in ImageJ software. Another experimental dataset was the data on CR3 receptor clustering on human neutrophils [2]. Detmers et al. experimentally obtained the distribution of CR3 clusters by their size after stimulation of polymorphonuclear neutrophils (PMN) by phorbol myristate acetate (PMA). For the parameter estimation for the “Clustering model,” we assumed that the maximal cluster size, *N*, equals 12 based on CR3 immunolocalization images from [2]. Model parameters were estimated on experimental data at time points 0 and 25 min, and the model was validated on data for time point 10 min from [2].

### 2.3. “2-Equation” Model

Despite being able to describe the experimental data accurately, the “Aggregation model” and the “Clustering model” are difficult to be used as parts of more complex models of multi-scale biological processes due to a large number of parameters and variables. Therefore, we constructed a reduced version of both models (a “2-equation” model). The reduction was based on the assumption that the main differences in the reaction rates are between the association/dissociation of a single particle (receptor) with an aggregate (cluster) and the association/dissociation of two aggregates (clusters). Therefore, we replaced all variables for clusters by a single variable, responsible for clusters of all sizes. The “2-equation” model has the same parameters as the more general “Aggregation model” because it aims to describe both aggregation and clustering processes. It consists only of two differential equations. These equations describe the behavior of single particles/receptors and the particle/receptor aggregates:(6)$$\{\begin{array}{c} \frac{dp}{dt}=-k_{1}np-2k_{2}p^{2}+k_{-1}n\\ \frac{dn}{dt}=k_{2}p^{2}-k_{-2}n^{2}+k_{3}n \end{array}$$
where *p* is the concentration of single particles, *n* is the concentration of aggregates/clusters, *k*_2_ is the probability of new aggregate/cluster formation from two single particles, *k*_1_ is the probability of another particle attachment to an existing aggregate/cluster, *k*_−1_ is the probability of single-particle detachment from an aggregate/cluster, *k*_−2_ is the probability of formation of one aggregate/cluster from two existing ones, and *k*_3_ is the probability of an aggregate/cluster fragmenting into two. The mean aggregate/cluster size, *s*, for “2-equation model” is calculated according to the formula:(7)$$s=\frac{p_{0}-p}{n}$$
where *p*_0_ denotes the initial concentration of particles. It should be noted here that such simplification of the models violates mass conservation law because the clusters, *n*, could have variable mass; that is why additional restrictions on *s* should be introduced in some cases.

The “2-equation model” implies the same mechanisms as previously published Smoluchowski-equation based models. For example, Becker–Döring model [43] also describes coagulation with dynamics of single particle attachment/detachment; however, it does not consider the possibility of larger particles/clusters detachment, which is included in the “2-equation model” (parameter *k*_3_). The main feature that distinguishes the “2-equation model” model from other published models is the description of all population of aggregates/clusters by a single variable.

### 2.4. Methods for Parameter Estimation, Model Solution, and Comparison of the Models

Each set of ordinary differential equations was integrated using the LSODA method [52,53] implemented in COPASI software and Python 3.6. Model parameters were assessed from experimental data by means of the following techniques implemented in COPASI software: Particle Swarm [54], random search, Hooke and Jeeves [55], or Levenberg–Marquardt [56]. Python 3.6 was used to investigate the properties of the “Aggregation model” and “Clustering model,” as they have a large number of differential equations.

We used Akaike’s Information Criterion (AIC) [57] to compare the “Aggregation model” and the “Clustering model” with the “2-equation model”. This is an information-theoretic criterion that is used to compare and select several models. It incorporates the cost function value from the parameter fitting process and also a penalty based on the number of parameters in the model. When applying the criterion, the best is considered the model that sufficiently fully describes the data with the least number of parameters. The criterion is defined by the AIC:(8)AIC=2K+n×ln(RSS)
where *K* is the number of parameters, *n* denotes the experimental sample size, and *RSS* denotes the residual sum of squares, which represents the total error of the approximation. The value of AIC has no meaning by itself and is used to compare models fitted to the same experimental data. The best model is the one with the lowest AIC criteria.

## 3. Results

### 3.1. Description of Protein Aggregation Data

In order to assess the ability of the constructed models to describe biological processes, we first assessed model parameters on one set of experimental data and then validated the model against another set of experimental data. Then, we performed an investigation of the impact of model parameters on its behavior and/or compared the ability to describe the experimental data between the full model (either “Aggregation model” or “Clustering model“) and the “2-equation model”.

Since aggregation is considered to be the most general case of coagulation and fragmentation, we started with the investigation of the “Aggregation model” and “2-equation model” models for the aggregation process and used the data of bovine serum albumin (BSA) aggregation.

We estimated parameters for the kinetics of BSA aggregation in the presence of different concentrations of ArgEE with fixed *p*_0_ = 1. Figure 2a shows that both models successfully described all sets of data. Parameters for each experimental curve appeared to be equal between both (“Aggregation” and “2-equation”) models (Table 1). Their values for different concentrations of ArgEE are given in Table 1. It appears that all parameter values reduced with the increase in the ArgEE concentration. Furthermore, from the comparison between relative *ki*-s changes, we can conclude that the suppressive effect of arginine derivatives is mainly caused by a decrease in particle attachment to an existing aggregate (Figure 2b).

### 3.2. Description of Platelet Aggregation Data

We estimated parameters for experimental OD-curves obtained for platelet activation by 2 μM and 3 μM of ADP. As shown in Figure 3a,b both models successfully described all sets of experimental data. The corresponding model parameter values are given in Appendix A. An example of parameter values for stimulation by 2 μM of ADP for “Aggregation model” are *k*_−1_ = 0.0001, *k*_1_ = 0.009, *k*_−2_ = 8.0 × 10^−7^, *k*_2_ = 0.0016, *k*_3_ = 0.0068, *p*_0_ = 4.8; and for “2-equation model” are *k*_−1_ = 0.0061, *k*_1_ = 0.0090, *k*_−2_ = 8.0 × 10^−7^, *k*_2_ = 0.0016, *k*_3_ = 0.0014, *p*_0_ = 4.24. The “Aggregation models” with maximal cluster sizes *N_max_* = 10 and *N_max_* = 100 describe experimental data with the same parameters and the same distribution of particles between aggregates of different sizes (Appendix A). Therefore, we can conclude that (1) the particles are mainly distributed between smaller aggregates, and (2) it is rational to use *N_max_* = 10 to approximate experimental data.

Furthermore, both models gave similar predictions on the kinetics of mean aggregate size and concentration of single platelets (Figure 3c,d). The models also predicted that the aggregate size achieves its maximum value earlier than the OD-curve achieves its minimum values, which is consistent with experimental data [33]. The “2-equation model” has the same approximation accuracy as “Aggregation model” (shown by Akaike’s Information Criterion (Appendix A)) and gives similar predictions; therefore, we assume that the “2-equation” model can be used instead of the “Aggregation model” in complex models.

### 3.3. Additional Restrictions on Parameter Values for “Aggregation Model”

The parameters described in Section 3.2 do not represent the unique solution for the particular set of experimental data. This could be seen from the fact that the parameters for the models describing OD-curves for 3 μM of ADP or 2 μM of ADP vary greatly (Appendix A, for example, *k*_−1_ equals 1.0 × 10^−4^ or 4.5 × 10^−17^). One way to improve the process of parameter estimation is to introduce additional approach for the description of the same experimental process and thus reduce the amount of variability in the model parameters. For example, the aggregation process can be described by the changes in the solution’s optical density or by the changes in mean aggregate size. We have used both experimental datasets (OD (*t*) and s (*t*)) simultaneously to obtain the parameter values for data our own experimental datasets on platelet aggregation (Appendix A). The new parameter values change a little between activation with ADP at 1 μM or 2 μM with the exception of *k*_−2_, which appears to be negligible. The parameters of single platelet attachment to aggregate appear to increase with ADP while the aggregate stability decreases in line with previously obtained results [33].

### 3.4. Description of Receptor Clustering Data

The parameters of the “Clustering model” were estimated from experimental data on GPVI receptor clustering on the surface of platelets [51] (as an example of a clustering process). As shown in Figure 4a,b, there are no significant differences between the approximations given by the “Clustering model” and the “2-equation model” for this dataset. Examples of parameter values for “Clustering model” (Figure 4a) are *k*_−1_ = 0.053, *k*_1_ = 6.1 × 10^−6^, *k*_−2_ = 10.2, *k*_2_ = 0.56, *p*_0_ = 10^4^ and for “2-equation model” are *k*_−1_ = 6.6 × 10^−9^, *k*_1_ = 2.1 × 10^−4^, *k*_−2_ =0.03, *k*_2_ = 7.5 × 10^−6^, *k*_3_ = 1.16, *p*_0_ = 10^4^. It should be noted that the parameter values between the “Clustering model” and the “2-equation model” are not comparable because they describe probabilities of different processes. The whole set of parameters for other experimental curves is given in Appendix A. The description of all sets of experimental data by the “Clustering model” and the “2-equation model” gave equal values of AIC. Therefore, these models describe the data similarly (Appendix A). Moreover, both models described the kinetics of single receptors concentration similarly (Figure 4c).

The ability of the “Clustering model” to describe the distribution of clusters by their size could be investigated by its application to the data on CR3 cluster distribution on the surface of human neutrophils described above. Parameter estimation was performed, as described in Section 2.2. The “Clustering model” accurately describes the kinetics of the mean cluster size at each point of time (Figure 5). The estimated parameters were as follows: *k*_1_ = 5.4 × 10^−5^, *k*_−1_ = 6.7 × 10^−4^, *k*_2_ = 3.4 × 10^−5^, *k*_−2_ = 6.7 × 10^−^^4^. The parameter values appear to be significantly lower for this model than for the model of GPVI receptor clustering, reflecting the longer characteristic times of the neutrophil activation compared to platelet activation. The parameters for “2-equation model” were also estimated and are the following: *k*_−1_ = 6.8 × 10^−10^, *k*_1_ = 0.0028, *k*_−2_ = 6.5 × 10^−4^, *k*_2_ = 3.2 × 10^−4^, *k*_3_ = 0.0061, *p*_0_ = 16.4 (Appendix A).

Therefore, we can conclude that the “2-equation” model can be used instead of the “Clustering model” in complex models.

### 3.5. Additional Restrictions on Parameter Values for “Clustering Model”

Analogous to the Section 3.3., additional restrictions are needed to ensure the correspondence of an experimental dataset to a single parameter set. Although no additional data could be obtained from the receptor clustering experiments, we derived the required restrictions from the energy conservation law and the physical properties of receptors. Firstly, the diffusion limits the clustering rates because the receptor interactions are limited by their mobility in the plasma membrane. At low values of Reynolds number, the mobility is proportional to 1/*r* (*r* is the particle radius) [58]. According to [59], the forward rate constants *k*_1_ and *k*_2_
~1r and these parameters are related to each other. Therefore, we can restrict the parameters values as [59]:(9)$$k_{2}=\frac{k_{1}}{2}$$

Secondly, to avoid the violation of the energy conservation law, we need to ensure that both monomer and dimer attachment/detachment does not alter the total free energy of the system [60]. This can be achieved by the following condition:(10)$$\frac{k_{2}}{k_{-2}}=\frac{k_{1}}{k_{-1}}$$

We have estimated the parameters for Col-III-induced GPVI clustering. The resulted estimation is presented in Appendix A.

### 3.6. Features of Receptor Clustering Process Revealed by the Mathematical Models

Based on the assumption that the “Clustering model” is more detailed than the “2-equation model” and may give insights into the mechanisms of the clustering process itself, we investigated its properties, i.e., steady states, characteristic features, and parameter dependencies.

First, we investigated the behavior of the mean cluster size. It appears that while some clusters may grow and then decrease in number, the overall pattern has an irreversible character, and eventually, the mean cluster size function reaches a steady-state (typical curve is shown in Figure 3a,b). Second, we looked for steady states. Although the steady-state of the system depends on the parameter values, three types of steady-state were observed, namely, (1) unclustered receptors when most of the receptors are single; (2) dimers, when most of the receptors are in dimers; and (3) full clustering, when most of the receptors are in clusters of maximal size (Figure 3d). Finally, the analysis of the impact of variation of parameters demonstrated that the steady states are determined not by the parameters themselves, but by their ratios *k*_2_/*k*_−2_ and *k*_1_/*k*_−1_; those represent dissociation constants of the corresponding reactions (Appendix A). The typical behavior of clusters of different sizes for each type of steady-state is presented in Appendix A. As could be expected, the dissociation rate values affect the steady-state of the system, while the parameter values affect the characteristic times to a steady-state.

The restriction introduced in Section 3.5 (Equation (10)) proposes that if receptor clustering is caused only by changes in the affinity of receptors to each other and does not consider any other mechanisms (e.g., cytoskeleton or ligand mediated clustering), then the resulting steady state can only be referred as type 1 or 3. With condition (Equation (10)), the mostly dimeric state (type 2) cannot be observed.

## 4. Discussion

In the present study, we developed three kinetic mathematical models for the description of particle aggregation and receptor clustering: (1) the “Aggregation model” to describe platelet aggregation, (2) the “Clustering model” to describe receptor clustering, and (3) the “2-equation model” capable of describing both processes.

While the “Aggregation model” and the “Clustering model” accurately describe experimental data with 10 variables, the “2-equation model” do this with only two variables. The simplicity of the “2-equation model” has several advantages and several drawbacks. For example, in comparison with the amyloid-β aggregation model presented by Ghosh et al. [61], which has more than 60 variables, the “2-equation model” could potentially describe the kinetics of aggregate formation while it is not capable of predicting the nucleation centers. Other models for protein aggregation [23,32] also contain restrictions on their parameter values that are applicable to their particular case but are not compatible with other aggregation cases. Since the “2-equation model” is not detailed, it cannot give specific information about the system, but as it resembles general behavior, this model could be used as part of more complex systems such as thrombus formation models [62,63], or it could be introduced in the Kinase–Phosphatase Segregation model of TCR activation to more accurately account clustering process [64] or into the mathematical models of platelet activation [65,66].

The novelty of the presented approach (the “2-equation model”) consists of the description of all aggregates/clusters by a single variable (*n*). This simplification does not lead to any loss in the model’s ability to describe experimental data of different types (monomer’s kinetics (Figure 2), mean cluster size (Figure 4), or optical density (Figure 3)). Additionally, in some cases, the values of the model parameters can give insights into the physical properties of the system (Figure 2b).

The performed investigation of the clustering model suggested that the cell has only three typical steady states, namely unclustered, dimers, and fully clustered (Figure 4d). This result suggests that there are not any other intermediate steady states and that cell tends to achieve the maximal local concentration of receptors to ensure high ligand binding efficiency, in line with experimental data [67].

The “Clustering model” may be considered as an extended Becker–Döring (BD) model. Indeed, it shares some common features, such as the fully clustered state achieved at some parameter values [68]. However, this state in the BD model is observed only for a particular set of parameters, and the number of parameters in these sets is unreasonably large, and it is complicated to account for all possible solutions. On the other hand, the “Clustering model” has only four parameters and is still capable of reflecting the main properties of the clustering process.

The parameters of the models presented here are derived by parameter estimation techniques and do not have a sound theoretical basis. The models presented here are general and do not take into account any particular properties of particular systems of interest. Therefore, the physical and biological nature of parameters varies considerably from system to system. For example, erythrocyte aggregation depends strongly on their collision rate [69], while this may not be correct for platelet aggregation [33]. Consequently, each system should be considered independently to obtain more detailed laws that govern each parameter. Development of a biophysical basis for the introduced parameters that could not only eliminate the uncertainty in the estimation but also give some additional information about the system will be the subject of further studies on particular cellular systems.

## 5. Conclusions

Here we developed a series of theoretical models for the processes of particle aggregation and receptor clustering. The simpler “2-equation model” was demonstrated to have the same estimation accuracy of experimental data as more extended and specific “Aggregation model” and “Clustering model”. Hence, it could be used as a simplification in the modelling of more complicated systems. Additionally, for the receptor clustering process, we demonstrated that the kinetics of the process make it possible to obtain the necessary, preassigned cluster distribution in steady or transient states.

## Figures and Tables

**Figure 1 life-10-00097-f001:**
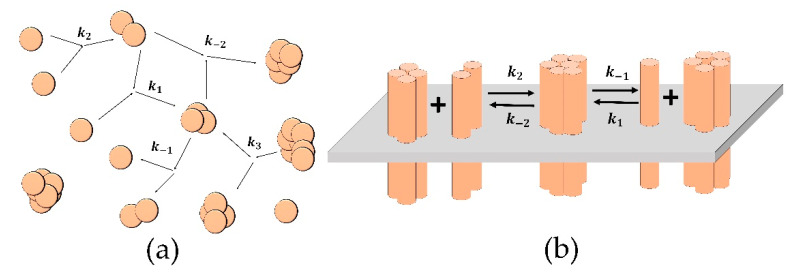
Schemes of the reactions for the “Aggregation model” (**a**) and the “Clustering model” (**b**). (**a**) The spheres represent single particles, and the merged spheres represent aggregates; the parameters of the reactions are: *k*_2_ is the probability of new aggregate formation from two single particles; *k*_1_ is the probability of another particle attachment to an existing aggregate; *k*_−1_ is the probability of the detachment of single-particle from an aggregate; *k_−_*_2_ is the probability of formation of one aggregate from two existing ones; *k*_3_ is the probability of an aggregate fragmenting into two. (**b**) The cylinders represent single receptors in the membrane (grey plane); the merged cylinders represent clusters; the parameters of the reactions are: *k*_1_ (*k*_−1_) is the probability of attachment (detachment) of single receptor to (from) a cluster; *k*_2_ (*k*_−2_) is the probability of attachment (detachment) of a dimer to (from) a cluster.

**Figure 2 life-10-00097-f002:**
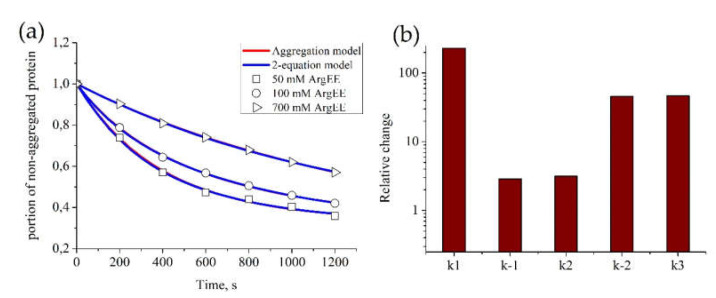
Parameter estimation for the “Aggregation model” and “2-equation model” based on BSA aggregation data. The estimation of model parameters for each model was conducted automatically. For each set of experimental data, parameters of the models were estimated independently. (**a**) Kinetics of BSA aggregation in the presence of ArgEE at the following concentrations: 50, 100, and 700 mM. Description of experimental data by the “Aggregation model” (red) or the “2-equation model” (blue); parameters of the models appear to be equal and are given in Table 1, as well as correlation of obtained parameters with ArgEE concentration. (**b**) Relative change between parameters, obtained for 50 and 700 mM of ArgEE.

**Figure 3 life-10-00097-f003:**
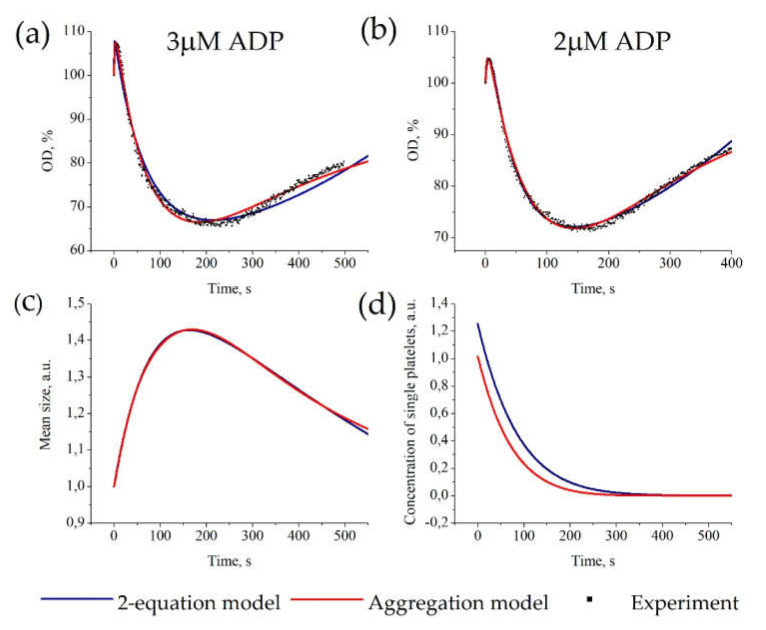
Parameter estimation for the “Aggregation model” and “2-equation model” based on platelet aggregometry data. Estimation of six model parameters and initial platelet concentration for each model was conducted automatically. For each set of experimental data, parameters of the models were estimated independently. Experimental data on platelet aggregation in response to 3 μM of ADP (**a**) or 2 μM of ADP (**b**). (**a**,**b**) Description of experimental OD-curves (dots) by the “Aggregation model” (red) or the “2-equation model” (blue); parameters of the models are given in Appendix A. (**c**,**d**) calculated time-course of the mean size of aggregate (**c**) and concentration of single platelets (**d**) for models describing experimental data for 3 μM of ADP [40,41].

**Figure 4 life-10-00097-f004:**
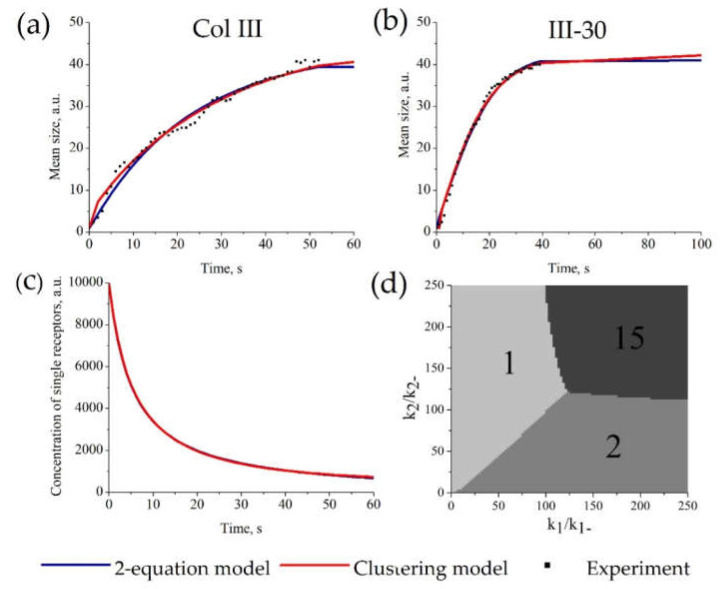
Parameter estimation for the “Clustering model” and “2-equation model” based on platelet GPVI receptor clustering. The estimation of model parameters and initial receptor concentration for each model was conducted automatically. For each set of experimental data, parameters of the models were estimated independently. Experimental data on GPVI receptor clustering on Col-III (**a**) or III-30 (**b**). (**a**,**b**) Description of the experimental amount of clusters (dots) by the “Clustering model” (red) or the “2-equation model” (blue); parameters of the models are given in Appendix A. (**c**) Calculated time-course of the concentration of single receptors (**c**) for stimulation with Col-III. (**d**) The size of predominant clusters in steady state. Typical *k*_2_/*k*_−2_ and *k*_1_/*k*_−1_ dependence of steady state for *N* = 15. There are only 3 types of steady states: unclustered, dimers, and full clustering.

**Figure 5 life-10-00097-f005:**
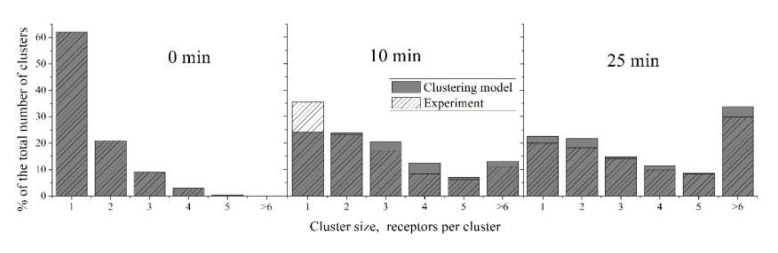
Description of size distribution of CR3 clusters on human neutrophils by the “Clustering model”. Experimental data [2] are shaded; model data are given in grey. Note that on the right panel, the sum of experimental data columns does not equal 100%. The parameters of the “Clustering model” were: *k*_1_ = 5.4 × 10^−5^, *k*_−1_ = 6.7 × 10^−4^, *k*_2_ = 3.4 × 10^−5^, *k*_−2_ = 6.7 × 10^−4^. The “Clustering model” adequately predicts the non-steady state at 10 min.

**Table 1 life-10-00097-t001:** Automatically assessed model parameters for experimental datasets given on Figure 2a.

Parameter	ArgEE Concentration, mM			Pearson Correlation Coefficient
	50	100	700	
*k* _1_	2.3 × 10^−3^	4.3 × 10^−3^	2.3 × 10^−3^	−0.69
*k* _−1_	1.3 × 10^−3^	1.6 × 10^−4^	4.5 × 10^−4^	−0.34
*k* _2_	8.6 × 10^−4^	7.1 × 10^−4^	2.7 × 10^−4^	−0.98
*k* _−2_	0.052	0.022	1.1 × 10^−3^	−0.85
*k* _3_	0.015	5.3 × 10^−3^	3.4 × 10^−4^	−0.80

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
