# Peer review of "Development of a Simple Kinetic Mathematical Model of Aggregation of Particles or Clustering of Receptors"

_life, 2020, doi:10.3390/life10060097_

Round 1

Reviewer 1 Report

The authors answered to all my comments.

Reviewer 2 Report

The authors have answered to all my questions and have improved the presentation of the paper significantly. I think that the paper is suitable for publication in Life now.  

This manuscript is a resubmission of an earlier submission. The following is a list of the peer review reports and author responses from that submission.

Round 1

Reviewer 1 Report

The authors propose a model for the aggregation of particles in biological contexts, particularly for the aggregation of particles, such as platelets, in 3 dimensions, and for the clustering of receptors on a membrane (2 dimensions). They develop two different models for these two cases: an “aggregation model” and a “clustering model”, respectively.

The models are based on the Smoluchowski aggregation-fragmentation equation (population balance equations), which is a system of ordinary differential equations (ODEs) with one equation for each of the sizes a cluster or aggregate can take. Therefore, it can be computationally expensive to evaluate and optimize such a model, particularly for large aggregate sizes (with respect to the monomer).  In order to avoid these problems, the authors also propose a simplified “2-equations” model that could capture the essential features of the process while largely reducing the computational cost of the solution. The reduced model consists of only two variables: single particles, monomers, and the rest of aggregates or clusters, reducing the size of the system of ODEs to only two equations.

Both models seem to be able to reproduce successfully experimental observations taken from different biological contexts. This leads the authors to conclude that their models, both the extended and the reduced one, can capture the dynamics of a wide variety of systems.

In general, the investigation is sound scientifically, as it is based in the well- established Smoluchowski equations which have been used to explain aggregation and fragmentation phenomena for the last century. Their models are able to fit the data in the experimental observations rather successfully. However, the authors do not seem to obtain much knowledge of the systems beyond these successful fits and do not draw any conclusion regarding the values that the parameters take. For example, for the platelet aggregation case, the value of the kernel for the case of fragmentation into a large aggregate and a monomer is 13 orders of magnitude smaller for 3µM ADP than for 2µM ADP (k1 = 4.5 × 10−4 for 3µM ADP and k1 = 10−17 for 2µM ADP), which seems to imply that either for greater concentration of ADP this kind of fragmentation is largely suppressed, or it could be a sign that the set of parameters to be extracted from the data could be improved. Expanding the discussion with respect to these results could improve the quality of the manuscript.

Finally, one conclusion, not at all trivial, is obtained from the clustering model.  The authors find that,  within the clustering model, there are only three distinct types of steady-states of the system: monomer-dominated, dimer- dominated or a system composed by the largest clusters possible. In particular, in my opinion, the authors could develop more on the causes and consequences that these interesting findings might have, as well as on the nature of the transitions between steady-states.

Nevertheless, there are some points I feel should be clarified or corrected prior to publication:

  • In eq. (1), I believe there is a typo, as there is a term fij,j[i + j] where it should be fij,j[i].
  • In eq. (2), I believe there is another typo, as for j = 2 there should be a gain term which is missing, proportional to k2  and stemming from monomer-monomer aggregation.
  • In eq. (4), what is the scaling parameter m? It does not appear in the rest of the manuscript so, is it necessary?
  • In Figure 2.c) what is the mathematical definition of mean size that was taken? In these arbitrary units, is the size of a monomer 1?
  • In Figure 4, for the 25 minutes distribution, why are the concentrations of all cluster sizes over-estimated? Should not it add up to 100% ? (I am not sure here if it is the experimental data that does not add up to 100% or is the theoretical prediction).
  • Regarding the choices of parameters to be used, I think it is important to explain why, in the clustering model, the rate of aggregation of a monomer and a dimer to form a trimer is the sum of two different rates (k1 and k2). Same for the fragmentation of a trimer into a monomer and a dimer (k1 and k2 ).
  • Regarding the “2-equation” model, I feel it is important to explain, mathematically, what are the approximations involved as that might help to explain the discrepancies in the parameters obtained for the extended and the simplified version. While on the main text for the aggregation model the parameters in between the complex and simplified models do not vary much, in Table S1, one can see huge differences in between the parameters in the two models.
  • Why do parameters change so much within the same system at different concentrations of ADP? Does it imply something about the system? (I am particularly thinking of the parameters that are negligible in one case, of order 1016 , that are not negligible in the other).

My recommendation is to accept the paper as long as the authors address the causes and consequences of the variability of the fitting parameters. While some other minor improvements should also be performed or discussed (see the bullet list), this is the only major obstacle for the publication of the manuscript in its present form.

Reviewer 2 Report

The present manuscript propose three theoretical models aimed at describing the kinetics is of aggregation and/or clustering of molecules in biological contexts, in particular for signaling. The models are not really new, even the one labeled « 2-equation ». There exists indeed in the physical chemistry community a large literature dealing with molecular self-assembly with similar models. The Becker-Doring model is just one particular example for clustering.

The authors claim that they found three models appropriate to describe various quantitative biological data, and that the « 2-equation » model is simple enough (as it has less parameters) to do the job of quantitatively describing datas. Well, this is not really surprising if one consider the literature mentioned. As a consequence, I personally did not find what was the new and original insight brought by this manuscript. Moreover, no biological or physical consequences are drawn from these quantitative fitting.

In its current state, I do not recommend publication of this manuscript, mainly because of these reasons (lack of originality, no consequences on the biological systems investigated). 

Reviewer 3 Report

In this paper, the authors develop mathematical models to describe the kinetics of 2D and 3D cluster formation.  They present three kinetics models, which they call 1. clustering, 2. Aggregation and 3. “2-equation model”.  The goal of the paper is to describe the experimental data on GPVI related to receptor clustering and platelet aggregation.  The authors conclude that the simplified models are enough to explain clustering in 2D and in 3D.

The authors have already published many of the results in this manuscript in Ref. 33.  Since the authors do not refer to Ref. [33] in the introduction, it is not obvious which parts of the current manuscript is novel and which parts have already been published in Ref. 33.  Since this is an extended version of Ref. 33, I would expect that the details of equations to be explained in more details and the results would be discussed more.  However, the paper is reference 33 is written more clearly and in fact has more details.

While I don’t recommend the publication of the paper in the current format, I find the subject of the paper and the results interesting. The authors need to completely rewrite the paper, significantly extend their mathematical modeling sections and discuss in more details the significance of their results. Here are more specific recommendations:

  1. The authors should explain what each term in Eq. 1 mean immediately after Eq. 1.
  2. The authors need to explain Beer Lambert law. Many readers are not familiar with the law. To this end, the origin of Eq. 3 should be explained.
  3. What is the relation between the lag time parameter and rate constants?
  4. What is the origin of m in Eq. 3? How do you justify it?
  5. In general, with so many parameters are your solutions unique? What methods have you used to test the uniqueness of the solutions? I know that compared to the other approaches, you have less fitting parameters. Nevertheless, with so many fitting parameters, one is able to fit anything.
  6. The English at the beginning of section 2.2.1. needs to be improved. The meaning of the paragraph is not clear.
  7. Eqs. 6 and 7 are already in ref. [33]. Are the authors calculating the same thing again?
  8. Can the authors explain Eq. 8 in more details? How accurate is it?
  9. At the beginning of the results section, the authors need to explain what experimental data they are trying to fit. References 39 and 40 are abstracts.
  10. While the authors state that they develop three models, these methods have already been presented by the authors in [33]. They need to clarify it.
